# Novel Permanent Magnetic Surface Work Hardening Process for 60/40 Brass

**DOI:** 10.3390/ma14216312

**Published:** 2021-10-22

**Authors:** Ayman M. Alaskari, Abdulaziz I. Albannai, Abdulkareem S. Aloraier, Meshal Y. Alawadhi, Tatiana Liptáková

**Affiliations:** 1Department of Manufacturing Engineering Technology, College of Technological Studies, PAAET, Shuwaikh 70654, Kuwait; ai.albannai@paaet.edu.kw (A.I.A.); as.aloraier@paaet.edu.kw (A.S.A.); my.alawadhi@paaet.edu.kw (M.Y.A.); 2Department of Material Engineering, Faculty of Mechanical Engineering, University of Zilina, 01026 Zilina, Slovakia; tatiana.liptakova@fstroj.uniza.sk

**Keywords:** surface work hardening, brass, permanent magnets, microhardness, martensitic stainless steel balls

## Abstract

Surface work hardening is a process of deforming a material surface using a thin layer. It hardens and strengthens the surface while keeping the core relatively soft and ductile to absorb stresses. This study introduces a permanent magnate surface work hardening under two opposite permanent poles of a magnet to investigate its influence on a brass surface. The gap between the brass and the north magnet pole—fixed in the spindle of a vertical machine—was filled with martensitic stainless steel balls. The rotational speed and feed rates were 500–1250 rpm and 6–14 mm min^−1^, respectively. The novel method improved the surface hardness for all parameters by up to 112%, in favor of high speed, and also increased yield by approximately 10% compared to ground samples. Surface roughness showed higher values for all speed–feed rate combinations compared to the ground sample. Nevertheless, it showed better roughness than other treated conditions with high and low feed rates. The ultimate tensile strength and ductility remained unchanged for all conditions other than the untreated brass. A factorial design and nonlinear regression analysis were performed to predict the microhardness equation and effectiveness of the independent variable—speed and feed rate—for the proposed process.

## 1. Introduction

Most failures in engineering components are initiated from the surface layer; therefore, surface treatments can play a crucial role in controlling the material performance and lifetime, as reported by Maleki et al. [1]. The demand for improving the wear resistance of workpieces, e.g., by surface hardening in industries, is an attractive choice because the wear of engineering components can cause a high clearance, along with precision loss, in dynamic structural parts [2]. Surface treatments modify surfaces of parts without affecting the core of the material. There are different engineering methods for the surface hardening of metals, such as layer additions and substrate treatment. The combination of a hard surface and resistance to breakage upon impact is useful in certain parts, such as a cam or ring gear, bearings or shafts, turbine applications, and automotive components, that must have a very hard surface to resist wear, along with a tough interior to resist the impact that occurs during operation [3]. In addition, the probability of crack initiation and propagation at the core of the workpiece can be reduced by applying compressive residual stresses to the surface [4].

Maleki et al. [1] studied several surface-severe plastic deformation techniques, including severe shot peening, laser shock peening, and ultrasonic nanocrystal surface modification; they investigated the effects of process parameters and the kinetic energy of each treatment on the microstructure, mechanical properties, and fatigue behavior of nickel-based superalloy 718. They found that among the applied treatments, ultrasonic nanocrystal surface modification is the most efficient in improving the mechanical properties because it leads to the most significant fatigue performance, followed by severe shot peening and laser shock peening. However, Soyama et al. [4] found that the surface roughness increases with conventional shot peening because local plastic deformation is required to introduce compressive residual stress and work hardening. On the other hand, Chen et al. [5] studied the effects of parameters of the ultrasonic shot peening (USP) process on grain refinement, surface hardness, and tensile properties of pure copper surface layers. They found that the grain size of the strengthening layer was less than 10 nm, the thickness of the grain-refined strengthening layer reached 338 μm, the hardness of the surface increased by 233.5%, and the tensile strength increased by 17.1% compared to that of the original pure copper specimens using USP stretching on both sides of the tensile specimens.

Other methods, such as burnishing, require improved surface roughness and sufficiently moderate surface hardening. Kalisz et al. [6] concluded that the burnishing process is more advantageous than polishing in terms of surface finish and tribological characteristics. Burnishing is not a chip-removal process. It depends on surface deformation, which can be performed simply by applying a highly hardened ball or roller subjected to external compression forces onto the surface of a flat or cylindrical workpiece. The ball or roller that is fed should be subjected to an appropriate direction according to the workpiece surface [7,8]. The applied force exceeds the yield stress of the material during the burnishing process; a distorted thin layer is formed on the surface, resulting in intensive plastic deformation. Hardening deformation occurs and results in a significant increase in layer hardness [9]. Hassan and Al-Bsharat [7] showed that the burnishing process parameters, such as forces and the number of passes, are vital parameters that affect the surface roughness of the workpiece during the burnishing process. They concluded that balls with larger diameters are more effective in improving surface roughness, while smaller-diameter balls are more effective in increasing surface hardness. They added that the surface roughness decreases to a certain limit with an increase in feed rate, burnishing speed, force, and the number of tool passes. Lou et al. [10] analyzed the impacts of burnishing process parameters, such as the number of passes, burnishing force, and ball diameter on the surface hardness and roughness of two different non-ferrous metals. They found that the burnishing force and number of passes are the most effective parameters for surface roughness and hardness. Moreover, Hassan [11] studied the influences of initial burnishing parameters on non-ferrous materials and showed that the initial surface hardness, surface roughness, burnishing ball diameter, and the application of different lubricants have a significant influence on the burnishing process. In addition, if the depth of cut increases, the burnishing force necessary to fill the gaps on the surface with protuberances also increases, thereby producing irregularities of the deformed metal surface [9]. The roughness and microhardness of the produced surface during the burnishing process are influenced by the ball diameter, burnishing force and the method of its application, burnishing speed, feed rate, number of passes, initial surface roughness, and hardness of the material to be processed [12,13]. Tripathi et al. [14] used the Taguchi approach and found that the rotational frequency, feed rate, force, and the number of passes have a significant effect on both surface hardness and roughness during brass burnishing. In addition, Rao et al. [15] showed that both roughness and surface hardness are improved by increasing the number of passes, with a maximum of four passes. Moreover, Morimoto [16] studied the effect of burnishing parameters, such as force, number of passes, and feed rate, on the work hardening of a turned mild steel at the burnished surface and subsurface. They concluded that the extent of work hardening of the workpiece surface gradually decreases with an increasing number of burnishing passes.

More advanced magnetic-aided burnishing processes have been developed since the 1980s using a ball, roller, and diamond burnishing [17,18,19]. Alaskari et al. [20] improved the surface quality, hardness, and corrosion resistance of 60/40 brass using a novel process, called flexible magnetic burnishing brush (FMBB), using cylindrical stainless steel pins as magnetic burnishing particles under only two permanent magnets. They showed that the burnishing speed and feed rate determine the surface quality and homogeneity, improving the microhardness by 40%, microroughness by 76%, and corrosion rate at a certain range of speed and feed rate. Alaskari et al. [21] examined the influences of multi passes and directions of FMBB on different initial brass surface conditions at a fixed optimum speed and feed rate. They proved that the initial surface roughness, number of passes, and reverse strain mechanism primarily affect the surface properties and integrity.

This study proposes a new technique to treat a metal surface by applying an oscillation compression load on the surface of the workpiece subjected to spherical stainless steel balls controlled by a strong moving permanent magnetic field at different speeds and feed rates. The purpose of the proposed technique, permanent magnate surface work hardening (MSWH), is to improve the surface work hardening of the material by increasing the surface yield strength and surface hardness and consequently increasing the surface roughness.

## 2. Experimental Section

### 2.1. Materials Preparation and Setup

Twenty 60/40 yellow brass (C274) plates with 90 mm length, 90 mm width, and 3 mm thickness were cold cut and then ground with 180 grit aluminum oxide. Two other plates with the same dimensions, but without initial grinding, were also cold cut to be used for the selection of optimum speed–feed rate combinations. All workpieces were rinsed and cleaned thoroughly with desalted water, cleaned with acetone, and dried. Martensitic stainless steel balls (grade 440C) 3/16 inches (4.76 mm) in size were used as magnetic hardening particles for the MSWH process. AISI 440C stainless steel has high strength, hardness, and wear resistance with moderate corrosion resistance. Table 1 lists the chemical compositions of the C274 yellow brass and 440C stainless steel balls.

### 2.2. Permanent Magnetic Surface Hardening Process

The universal milling machine equipped with a frequency converter was used in all burnishing processes in this study. The frequency converter was used to reduce the minimum allowable feed rate of the saddle (milling table) of the machine up to a minimum of 2.8 mm/min. The N52 cylindrical-type permanent magnet (N52) with a 25 mm diameter was fixed inside the collet chuck of the milling machine, where the north pole of the magnet faced the milling table. The N52 magnet is the strongest commercially available permanent rare earth magnet; it is made of iron, boron, and neodymium. A South N52 magnetic pole with the length, width, thickness of 90 mm, 90 mm, and 10 mm, respectively, was placed on a vice that was fixed on the machine table and located below the yellow brass workpiece.

An average of 30 martensitic stainless steel balls were placed between the yellow brass and the north pole magnet (Figure 1) with varying burnishing speeds (500, 750, 1000, and 1250 rpm) and feed rates (6, 8, 10, 12, and 14 mm/min). These 20 speed–feed combinations were processed without lubrication. Two other plates with the same dimensions and cut conditions, but without initial grinding, were hardened using a selected high speed and low feed rate to obtain the optimized speed–feed rate combination. These two speed-feed rate combinations were 1500 rpm–4 mm min^−1^ and 1750 rpm–2.8 mm min^−^^1^. The maximum speed, 1750 rpm, was limited to the retention of the balls in the magnetic field during the proposed process. The minimum feed rate, 2.8 mm min^−1^, was used due to the limitation of the frequency converter.

### 2.3. Experimental Tests

Vickers microhardness (HV) of the workpiece surface with a force of HV 0.1 was evaluated using an Innovatest 400 series testing machine (INNOVATEST, Maastricht, The Netherlands). Six indentations with a 3 mm gap between each indentation were considered, starting from the center of the surface hardening to the unhardened surface. The ASTM E384 standard [22] was followed for all the workpieces.

The top surfaces of the workpieces were thoroughly cleaned, evaluated, and compared with the unhardened surfaces using an optical microscope (ZEISS. Oberkochen, Germany), as shown in Figure 2. These evaluations were conducted 15 mm away from the centerline of the hardened surface, where the width of the hardened surface was approximately 40 mm. At the same location, cross-sectional surfaces were evaluated and compared with the unhardened surface using ZEISS optical microscope (for cross-sections) after etching with ferric chloride. These surfaces are shown in Figure 3; the affected depth of surface hardening was 72 µm, as measured by ImageJ software (bundled with 64-bit Java 1.8.0_172).

In addition, the ultimate tensile strength, yield, and ductility for all workpieces, including only ground conditions, were evaluated for the ground surface using tension testing (Tinius Olsen, H100KU, Horsham, PA, USA). The tensile samples were prepared based on ASTM E8/ E8M-21 [23] with a gauge length and width of 25 mm and 6 mm, respectively. Moreover, the surface roughness of all samples was evaluated using a roughness test (MarSurf PS 10 from Mahr, Providence, RI, USA) for 6 mm in the same direction as the feed.

## 3. Results and Discussion

The application of a high compressive force generated by a permanent magnetic field between the north and south poles caused oscillation of the martensitic stainless steel balls on the workpiece surface, resulting in high surface plastic deformation of the brass. This result agrees with observations presented by Dzierwa and Markopoulos [24]. In the present work, surface work hardening was performed using a novel method by employing permanent magnets whose two magnetic poles were fixed on a vertical milling machine. The hardening was realized by martensitic stainless steel balls moving under the brush being kept by a magnetic field. In the presence of the magnetic field, the relative motion between these balls and the brass workpiece, generated from both rotation speed and feed rate, caused plastic deformation of the surface using ball oscillation and friction processes. The rotational speed and feed rate were the main variable parameters of the process, while the other parameters such as the gap between the spindle and the workpiece surface, diameter of the rotating magnetic pole, and density of the magnetic balls were all kept constant. The surface characteristics of the hardened samples were affected by the relative motion and force of the magnetic brush. 

### 3.1. Optical Microscope

The plastic deformation was clearly observed from the microscopic side-view figures of the unhardened and hardened surfaces (Figure 2a,b). The unhardened brass surface (Figure 2a) had a straight uniform surface with an equal-sized grain. For the hardened sample, the surface was marked by a martensitic stainless steel ball, causing high local plastic deformation of the grain on the brass surface. The affected plastic zone was located between the maximum plastic strain and zero-plastic strain, as shown in Figure 2b; grain refining was located on the surface at a depth of 72 µm. Therefore, the MSWH process plastically deformed the surface of the workpiece with forces driven from the magnetic field in the presence of both speed and feed rate. These results agree with cross-sectional optical micrographs of Dai [25], showing a highly deformed layer on the surface of pure titanium processed by high-energy shot peening. It also agrees with the grain refinement observed on the surface layer of ANSI 304 stainless steel deformed by laser shock processing [26].

Figure 3 presents a top view of the sample surface before and after hardening with MSWH at specific speed–feed rate combinations. Indentation and friction marks on the brass surface created due to the relative speeds of the stainless steel balls provide clear visual evidence of the effect of the proposed surface work hardening process. We notice that the surface hardening for low feed rates (Figure 3a,c) have a more marked intensity of hardening indentations than high feed rates (see Figure 3b,d). Figure 2 and Figure 3 demonstrate the MSWH-generated forces that are sufficiently high to deform the brass surface plastically.

### 3.2. Microhardness

The HV values of the test samples were evaluated. The results show that the average microhardness is significantly improved for all speed–feed rate combinations (Figure 4a). They increase surface microhardness from 42.7 HV (33%) to 144.3 HV (112%) based on the high speed and mostly high feed rate, as shown in Figure 4b. Only one exception occurred at a speed of 1250 rpm, where the highest microhardness average was observed at a low feed rate of 6 mm min^−1^; it also exerted the optimum condition among all the tested conditions. The results of this study conform to those of previous studies that have reported an increase in hardness values on the deformed layer of materials treated by shot peening [27], the waterjet peening process [28], ultrasonic cavitation modification [29], flexible magnetic burnishing brush [20,21], and the ultrasonic surface rolling process [30].

The high relative speed between the stainless steel balls and workpiece—created by a combination of speed and feed rate—significantly increased the compressive stresses on the surface, increasing its hardness. The high relative speed mostly originated from the high rotational speed of the north pole magnet fixed in the spindle. At high rotational speeds, the increase in the feed rate increased the relative speed of the balls until it reached a point where it was difficult to hold the stainless steel balls in the process area. Additionally, the high relative speed was still valued at high speed and low feed rate; however, the balls repeatedly deformed the same area, increasing the hardness values. The microscopic side view shown in Figure 2 indicates the presence of a gradient of microhardness values where the highest value is at the top layer; the value decreases gradually toward the depth of the sample. This is similar to previous studies that reported a gradient microstructure in Cu-Al alloy after surface strengthening [31,32], shot peening of TA17 titanium alloy [33], and surface mechanical rolling treatment of AISI 316L stainless steel [34].

Univariate analysis of variance and nonlinear regression analysis were performed using SPSS software (SigmaPlot 12.0) to evaluate microhardness and to predicate the effectiveness of the independent variable (speed and feed rate); the outcomes can be seen in Table 2 and Table 3, respectively. A factorial design was used in the univariate analysis of variance using four levels of hardening speed (500, 750, 1000, and 1250 rpm) and five levels of feed rate (6, 8, 10, 12, and 14 mm min^−1^), with six repetitions for each case. Table 2 shows that the hardening speed, feed rate, and their interactions significantly affect the microhardness (measured in HV). Table 3 shows the nonlinear prediction of microhardness; the equation can be written as
*H =* 64.283 × *S*
^0.159^ × *f*
^0.113^(1)
where H represents the microhardness of MSWH measured in HV, S represents rotational speed in rpm, and f represents feed rate in mm min^−1^.

Based on Equation (1), microhardness increases with increasing speed and feed rate, but with more weight in favor of the speed because it has a higher power value of 0.159. This also agrees with the results in Figure 4, where the highest microhardness values among each feed rate are recorded at the highest speed of 1250 rpm. A high speed provides a process with high compressive forces that lead to high surface work hardening. However, a high feed rate affects microhardness values, specifically when combined with high speed. Therefore, the microhardness values increase as the relative speed between the stainless steel balls and brass increases. This can be obtained by a higher speed and high feed rate if the stainless steel balls remain in the magnetic field during the process. The relatively high speed of the balls increases the magnetic pressure, increasing the plastic deformation on the surface of the tested workpiece.

For two initial non-ground samples representing speed–feed rate combinations of (1500 rpm–4 mm min^−1^) and (1750 rpm–2.8 mm min^−1^), the microhardness values increased to 275.7 HV (114.9%) and 286 HV (123%), respectively. Therefore, the values of microhardness are primarily affected by the high speed, where the combination of speed and feed rate affects the relative motions and retains the balls in the magnetic field.

### 3.3. Mechanical Properties and Surface Roughness

The values of both ultimate tensile strength (UTS) and ductility obtained from tension testing, including the ground brass, remained unchanged at 380–385 MPa and 67.2%–71.9%, respectively. However, Figure 5 shows that the yield strength of all samples processed with MSWH increases by 20.3–27.3 MPa compared to ground samples, irrespective of the applied speed and feed rate. The UTS and ductility of the brass were not affected by the surface-hardened MSWH because the process only affected the surface at a depth of 72 µm (see Figure 2b). The hardened depth did not contribute to the change in the overall UTS and ductility. These properties depend mainly on the total cross-section of the material and not on the thin surface conditions. However, such affected depth increases the yield strength up to only 27.4 MPa (10.8%), contrary to an over 144.3 HV (112.5%) increase in surface microhardness. Hence, the overall material strength and ductility remain unchanged, if not insignificantly improved, while the surface microhardness significantly improves, meeting the criteria of the surface work hardening mechanism. The current results are similar to the results reported for Cu-Al alloys treated by a surface strengthening mechanism, where an increase in the yield strength was noticeable; however, there was a minor change in the UTS associated with a slight decrease in the ductility [31].

The root mean square height (Sq) and arithmetic mean height (Sa) were evaluated for the ground and surface-hardened samples, as shown in Figure 6. The results show that Sq is higher for all speed–feed rate combinations than for the ground sample. The Sq values of the ground and the optimum surface-hardened samples were 10.5 µm and 27 µm, respectively. The Sa values for the ground and best surface hardening condition samples were 1.1 µm and 3.3 µm, respectively. The high values of roughness are due to the high indentation marks of the stainless steel balls on the brass surface, oscillations of balls, and friction, causing its surface to plastically deform, as shown in Figure 3. We notice that surface roughness is not affected mainly by rotational speed, while roughness at both low and high feed rates (6 and 14 mm min^−1^) shows better Sa and Sq values than the other feed rates. At a low feed rate, more surface area is work-hardened with compressive forces exerted by the balls in the MSWH process. However, at a high feed rate, more friction effects are exerted on the treated surface. Therefore, the feed rate and its combinations with the processing speed contribute to changing the surface roughness.

## 4. Conclusions

This paper demonstrates a novel MSWH process with stainless steel balls driven by permanent magnetic poles to improve the surface hardness of C274. The experimental results demonstrated the several effects of different factors on the surface characteristics, as follows:The MSWH process plastically deforms the brass surface to a depth of 72 µm via the oscillation of balls and friction processes with forces originating from the permanent magnetic field and in the presence of both speed and feed rate.The relatively high speed between the stainless steel balls significantly increases the compressive stresses on the surface; therefore, it increases the hardness by 42.7 HV (33%) to 144.3 HV (112.5%) depending on the high speed and primarily the high feed rate. The increment can be improved up to 157.7 HV (123%) with a proper speed–feed rate combination.Factorial design was performed using univariate analysis of variance using four levels of hardening speeds and five levels of feed rates, with six repetitions for each case. The results show that hardening speed, feed rate, and their interactions significantly affect microhardness value.A nonlinear prediction of microhardness using Equation (1) shows that microhardness increases as speed and feed rate increase, but with more weight in favor of the speed.If the balls remain in the magnetic field during the process, the magnetic pressure increases as the relative speed between the stainless steel balls and brass increases. It then increases the plastic deformation on the surface of the brass.Both the UTS and ductility remain unchanged, but the yield strength for all processed samples increases by 27.3 MPa (10.8%) regardless of the applied speed and feed rate. The hardened depth of 72 µm, for 1500 rpm and 6 mm min^−1^, could not contribute to changes in the overall UTS and ductility, while it contributed to increasing the yield strength.Using MSWH, Sq and Sa increase because of the high indentation marks of the stainless steel balls on the brass surface in low and high feed rates and their combinations.

## Figures and Tables

**Figure 1 materials-14-06312-f001:**
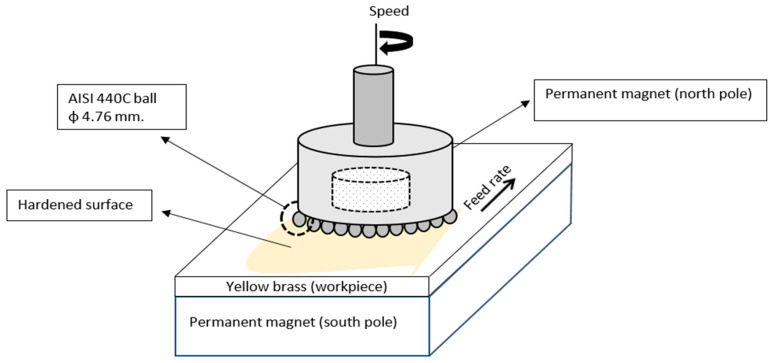
Magnetic surface work hardening process performed in this work.

**Figure 2 materials-14-06312-f002:**
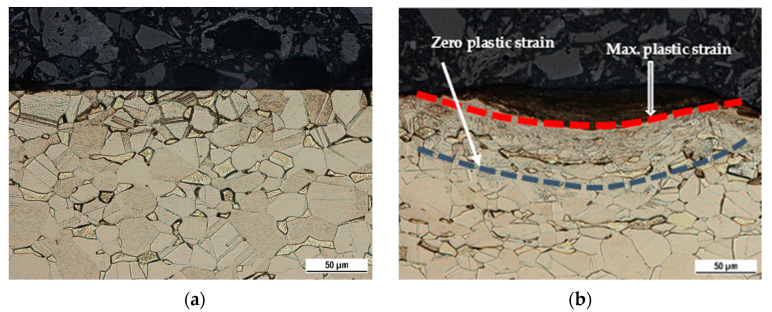
Cross-sectional metallographs of surfaces. (**a**) Unhardened surface; (**b**) at 1500 rpm and 6 mm min^−1^.

**Figure 3 materials-14-06312-f003:**
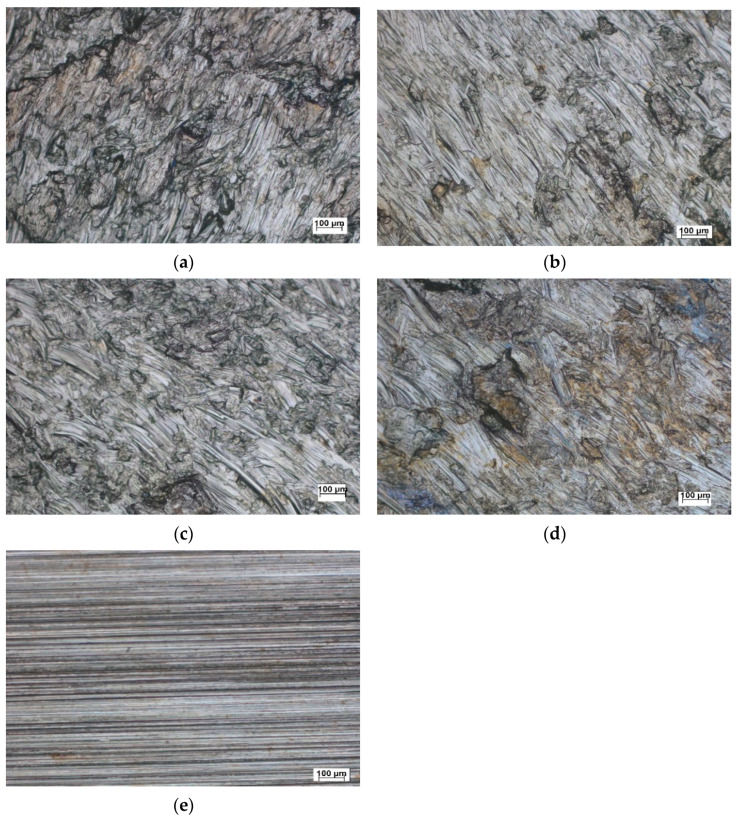
Topography of some speed–feed rate MSWH and ground surfaces: (**a**) 500 rpm and 6 mm min^−1^; (**b**) 500 rpm and 14 mm min^−1^; (**c**) 1250 rpm and 6 mm min^−1^; (**d**) 1250 rpm and 14 mm min^−1^; (**e**) ground sample.

**Figure 4 materials-14-06312-f004:**
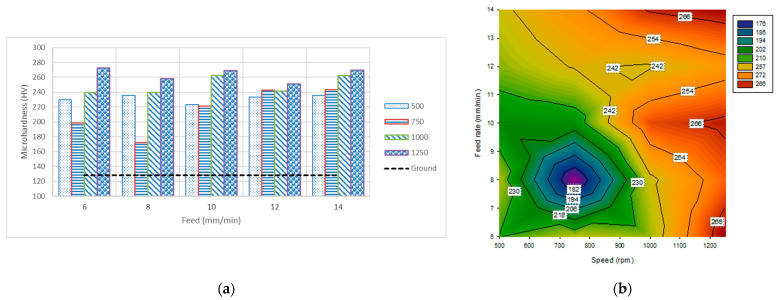
Average values of microhardness for various hardening speeds and feed rates. (**a**) Bar chart and (**b**) contour.

**Figure 5 materials-14-06312-f005:**
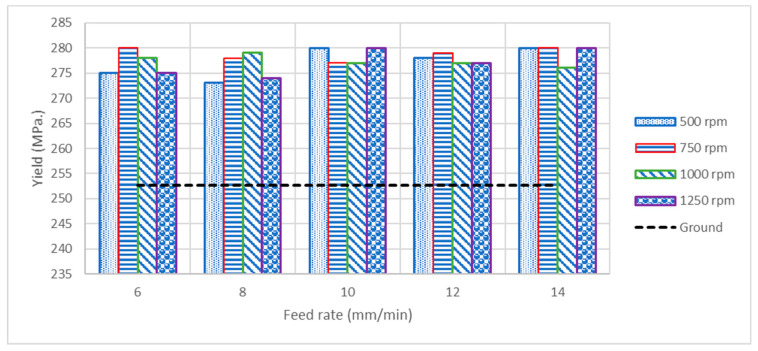
Bar chart of yield strength for various hardening speeds and feed rates.

**Figure 6 materials-14-06312-f006:**
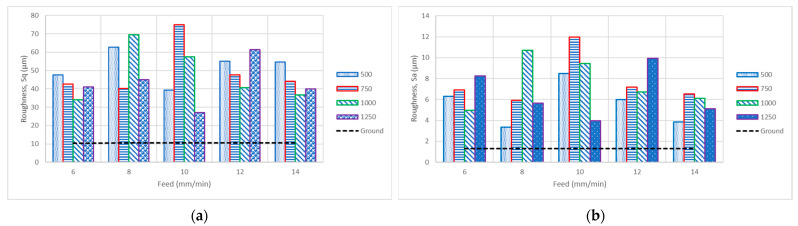
Bar chart of surface roughness for various hardening speeds and feed rates. (**a**) Root mean square height (Sq) and (**b**) arithmetic mean height (Sa).

**Table 1 materials-14-06312-t001:** Chemical compositions (wt.%) of yellow brass workpieces and stainless steel spheres.

Material	Fe	Cr	Ni	Mn	Cu	Mo	Zn	C
C274 (yellow brass)	-	-	0.13	-	60.68	-	39.19	-
SS 440C (balls)	79.15	17.05	-	0.95	-	0.72	-	1.09

**Table 2 materials-14-06312-t002:** SPSS outcome for univariate analysis of variance of microhardness.

Tests of Between-Subjects Effects					
Dependent Variable: Microhardness					
Source	Type III Sum of Squares	df	Mean Square	F	Sig.
Corrected Model	69,754.886	19	3671.310	52.683	0.000
Intercept	6,978,099.677	1	6,978,099.677	100,135.927	0.000
Speed	35,326.914	3	11,775.638	168.981	0.000
Feed	12,969.265	4	3242.316	46.527	0.000
Speed × Feed	21,458.707	12	1788.226	25.661	0.000
Error	6968.627	100	69.686	-	-
Total	7,054,823.191	120	-	-	-
Corrected Total	76,723.514	119	-	-	-

**Table 3 materials-14-06312-t003:** SPSS outcome for nonlinear regression analysis to predict the microhardness of MSWH.

**Iteration History**				
**Iteration** **Number**	**Residual Sum of Squares**	**Parameter**		
		**a**	**b**	**c**
1.0	7,054,823.191	0.000	0.000	0.000
1.1	6,536,501.331	9.129	0.000	0.000
2.0	6,536,501.331	9.129	0.000	0.000
2.1	322,848.497	19.678	0.172	0.505
3.0	322,848.497	19.678	0.172	0.505
3.1	126,846.409	22.166	0.190	0.480
4.0	126,846.409	22.166	0.190	0.480
4.1	91,876.500	25.297	0.205	0.374
5.0	91,876.500	25.297	0.205	0.374
5.1	79,756.510	35.299	0.218	0.173
6.0	79,756.510	35.299	0.218	0.173
6.1	67,914.867	48.136	0.190	0.123
7.0	67,914.867	48.136	0.190	0.123
7.1	54,908.458	61.009	0.162	0.113
8.0	54,908.458	61.009	0.162	0.113
8.1	49,148.869	64.230	0.159	0.113
9.0	49,148.869	64.230	0.159	0.113
9.1	49,142.959	64.283	0.159	0.113
10.0	49,142.959	64.283	0.159	0.113
10.1	49,142.959	64.283	0.159	0.113
**Parameter Estimates**				
			**95% Confidence Interval**	
**Parameter**	**Estimate**	**Std. Error**	**Lower Bound**	**Upper Bound**
a	64.283	10.747	42.999	85.568
b	0.159	0.023	0.113	0.204
c	0.113	0.026	0.061	0.164

## Data Availability

Data is contained within the article and can be requested from the corresponding author.

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
