# Peer review of "Novel Permanent Magnetic Surface Work Hardening Process for 60/40 Brass"

_materials, 2021, doi:10.3390/ma14216312_

Round 1

Reviewer 1 Report

Dear authors, it was very interesting to read your manuscript. But aspects arise and questions remain, the revision or answering of which would improve the quality of the manuscript.

  • In many cases you present improvement of properties by a percentage. But normally readers are interested in absolut values. In the opinion of the reviewer you should primarily indicate the absolute value achieved and, if you wish, also the percentage value.
  • You are talking about an frequency inventor. Are you sure that this wording is correct?
  • Please add how many balls were used in an experiment
  • At which position on the sample the process started and at which position it ends?
  • Can you characterize the path of a ball or is it a random path?
  • If possible, please add information about the average number of balls that passes a position of the samples.
  • Please add information about the number of balls that leave the magnetic field during an experiment. There seems to be an dependence to the process parameters.
  • Sometimes you are writing 1500-4 rpm and mm min-1 . You should never separate a number from its unit.
  •  In the text you are writing that workpiece and magnetic south pole have the same length and width. This should be taken into account in Figure 1, too
  • The reviewer has a problem in understanding the exact position of hardness and roughness measurements, sampling and orientation of the micro graphs and the taking of the tensile specimens. A supplementary sketch is necessary here.
  • Line 153: a force of 0.1 kgf makes no sense. The unit is N. You can add the force devided by 9,81 to HV. For example if the test load was 98 N you can write HV10. May be in your case HV 0.1?
  • Line 165: Image j is not given.
  • In the opinion of the reviewer a separation of Figure 3 from the roughness measurements is not the optimal solution.
  • Line 180: Are you sure that you have measured the surface roughness by use of a hardness test?
  • Figure 4a: You present the average values of 6 measurements for each variant. Whats about the scattering? You should offer the readers an idea of these results.
  • Equation 1: Please add information why you chose  this multiplicative approach from power functions.
  • Figure 4b: The contour plot of hardness as function of the process parameters looks very nice. But you should add how it was calculated. Did you use equation 1?
  • The abbreviation PMSH was not defined up to line 252.
  • In the opinion of the reviewer tables 2 and 3 are only understandable by specialists. Instead of these tables the authors could add the average difference between measurements and model and the estimates and std. errors of the model parameters.
  • Line 305 - 306: The reviewer cannot agree with the statement "surface roughness is not affected by rotational speed". The measurements show another result!
  • Conclusions: Is it correct that all investigations result in a plastic deformation zone of 72 µm thickness?

Author Response

Dear Reviewer

I, along with my coauthors, would like to thank the reviewers for their valuable comments and to re-submit the attached manuscript entitled “Novel Permanent Magnetic Surface Work Hardening Process For 60/40 Brass” (Manuscript ID: materials-1401517).  Please see the attached file.

Reviewer 2 Report

The article is devoted to surface work hardening by the use of the novel method employing permanent magnets. The hardening, in fact, is realized by martensitic stainless steel balls moving under the brush being kept by a magnetic field.

The introductory part is very correctly written. The authors have presented a comprehensive literature review concerning surface hardening in materials. It is one of the most thorough and reliable state-of-the-art I’ve ever seen. The last and the only paragraph of the introduction describes the novelty of the work and the scope. In my opinion, however, it is enough for the clear presentation of what the authors did beyond state of the art.

The experimental part is written clearly and correctly. However, I would ask the authors to correct figure 1: it’s a very helpful schema but prepared in an unprofessional way – the view perspective is incorrect. Maybe authors could ask a computer graphic designer for help? In my opinion, the article is worthy of better visualization of the method. The rest of the pictures are correct.

I have no objections to the section results and discussions. All is described correctly. Also, conclusions are well-written. I like itemization – such a form makes all the conclusions clear.

Summing up, the article is well-written and scientifically valuable. I recommend acceptance it after minor correction – redrawing figure 1.

Author Response

(The authors gave the same response as above.)

Reviewer 3 Report

Dear Authors:

The topic and idea were very interesting.  However, the description was not enough to assure me of the contents, unfortunately.  The reason may be attributed to the novelty of the process.  However, The following points would be beneficial for readers to follow the interesting results precisely. 

(a) The discussion part should be separated from the experimental result part.

(b) Then the mechanism for your process should be described and discussed in the discussion part. 

(c) Would you mind clarifying the role of the magnetic field in detail? 

(d) The effect of the magnetic field strength, the configuration of magnetic setting should be discussed more.

(e) Why did you use yellow brass for the experiment.  Please describe the reason, the need, and the applicability to other materials. 

Author Response

(The authors gave the same response as above.)

Round 2

Reviewer 3 Report

Dear Authors: 

I appreciate your sincere effort for the revision.  Hopefully, the manuscript will attract lots of readers.